# Clinical & laboratory profiles and treatment outcome of Kawasaki disease in children: Experience from a tertiary care hospital

**Shareen Khan**[1]*, **Md Abid Hossain Mollah**[1], **Abdul Baki**[1], **Nabila Tabassum**[2], **Nasreen Islam**[1], **Amrita Lal Halder**[1], **Tasnima Ahmed**[1], **Nurun Nahar Fatema**[3], **Jebun Nahar**[1], **Fauzia Mohsin**[1], **Mohammod Jobayer Chisti**[4]

1 Faculties, Department of Paediatrics & Neonatology, BIRDEM General hospital (Mother & Child wing), Segunbagicha, Dhaka, Bangladesh, 2 Pediatric Speciality Trainee, Department of Pediatrics, Nottingham University Hospital, United Kingdom, 3 Retd. Ex Head of the Department, Paediatrics and Paediatric Cardiology, Armed forces medical college and Combined Military Hospital, Dhaka, Bangladesh, 4 Dhaka Hospital, Nutrition Research Division, International Centre for Diarrhoeal Disease research, Bangladesh, Dhaka, Bangladesh

* shareenkhan26@gmail.com

## Abstract

### Background

A total of 40 cases of Kawasaki Disease (KD), diagnosed & categorized on the basis of criteria set by the American Heart Association were studied. We sought to evaluate their clinical and laboratory analysis along with their treatment.

### Methods

After inclusion, they underwent thorough clinical and laboratory analysis and were treated with intravenous immunoglobulin and aspirin. Their clinical & laboratory findings and treatment outcome were analyzed further using Epi info and at a probability of (p value) < 0.05, was considered statistically significant.

### Results

The mean age of the cases was 4.1 ± 2.9 years and boys outnumber the girls (57% vs. 43%). Most (65%) of the cases were incomplete KD compared to 35% as complete KD. The mean duration of fever was 11.7 ± 5.53 days. The frequency of the different classic clinical findings of KD were- cracking of lips & redness of oral cavity (72%), polymorphous skin rash (50%), conjunctival injection (45%), changes in the distal extremities (38%), cervical lymphadenitis (30%) and erythema on BCG scar (7.5%). High C-Reactive protein (CRP > 10 mg/dl), thrombocytosis, anaemia and leukocytosis were 85%, 45%, 43%, 38% respectively. Echocardiography showed coronary artery abnormalities (CAA) dilatation and aneurysm in 82.5% and 12.5% cases respectively which came down to 68% and 10% after receiving treatment with IVIg and Aspirin.

**Data availability statement:** Data cannot be shared publicly because of the fact that the data can only be shared publicly with the permission of Ethics Committee of BIRDEM. Data are available from the BIRDEM Institutional Ethics Committee (contact via professorahm@yahoo.com) for researchers who meet the criteria for access to confidential data.

**Funding:** The author(s) received no specific funding for this work.

**Competing interests:** The authors have declared that no competing interests exist.

## Conclusion & Recommendation

Diagnosis of KD is essentially clinical and criteria-based and 95% were complicated with CAA. Timely intervention with IVIG was mostly effective. Therefore, high index of suspicion of a long-continued febrile child is recommended to reach the diagnosis and to initiate timely intervention so as to potentially prevent CAA.

## Introduction

Kawasaki disease (KD) is an acute febrile illness of childhood seen worldwide and highest among the Asians [1]. The aetiology of KD is unknown but certain genetic markers (such as HLA-B51 and HLA-Bw22j2 serotypes, chemokine receptor gene-cluster CCR2-CCR5 haplotypes and FCGR3A polymorphism of the IgG receptor IIIa) show a predisposition to the disease. Clinical and epidemiological patterns suggest that it follows exposure to infectious agents like human coronaviruses, bacteria, fungi, house mites but to date, no single agent has been identified as the leading cause [2,3]. Clinical features are characterized by fever lasting for at least five days along with changes in the oral cavity, erythema of hands and feet, bilateral non-purulent conjunctivitis, polymorphous skin rash and cervical lymphadenopathy [1]. Additional symptoms include gastrointestinal symptoms, aseptic meningitis, mild hepatitis, urethritis, uveitis, sterile pyuria and arthritis. Patients vaccinated with BCG vaccine may show reactivation of the inoculation site. KD shock syndrome is a rare but severe form of the illness in which patients present with vasodilatory shock, hypotension, and poor perfusion, with or without myocardial dysfunction [4]. Kawasaki disease is an acute inflammatory vasculitis of medium sized elastic arteries that has a high predilection for the coronary arteries [5]. It is the leading cause of acquired heart disease in children and approximately 20−25% of untreated and less than 5% of the treated KD patients develop coronary artery abnormalities (CAA) [1]. The initial treatment of KD includes single dose of IVIg and first at a moderate anti thrombotic dose till the patient is afebrile followed by a low antiplatelet dose [1].However, children with high risk (Z score ≥ 2.5 on initial echocardiogram and age < 6 months) may benefit from intensification of therapy with IVIg plus adjunctive anti-inflammatory therapy (corticosteroids, tumor necrosis factor α inhibitors (e.g., infliximab and etanercept,interleukin-1 inhibitor, e.g., anakinra, cyclosporine) to reduce the risk of coronary artery aneurysm. IVIg resistance KD (persistent or recrudescent of fever at least 36 hours and < 7 days after completion of 1st IVIg infusion) are treated with 2nd dose of IVIg or IVIg plus methylprednisolone, infliximab, Anakinra, Cyclosporine, Cyclophosphamide, plasma exchange [4].

The incidence of KD has been increasing rapidly globally. In North America, Europe and Australia, the incidence is 5–22 per 100,000 children <5 years old [6]. Countries like Japan, Korea and Taiwan report an incidence of more than 10 times greater than North America, Australia and Europe [7]. In countries like India, China, Srilanka, the incidence of KD has been gradually increasing. Before 1990, there were only three reported cases in almost all regions of India but after that it has

been reported in almost all cities of India with 4.5 cases per 100,000 children less than 15 years. In China, the reported incidence varies between 7.06 & 55.1 per 100,000 children < 5 years [8]. Although, a study from the Department of Paediatrics, Bangabandhu Sheikh Mujib Medical University (BSMMU) in Bangladesh showed that there was an increase in the number of KD cases over the last 5 years, the exact evidential incidence of KD in this country is still unknown [9]. Therefore, the objective of the study was to see the clinical & laboratory profile of KD patients and their response to treatment.

## Rationale of the study

Considering the high incidence of acquired heart disease (coronary artery abnormalities (CAA) among untreated children with KD, awareness of the physicians/ pediatricians on case detection of this growing problem will help non-missing of the cases and thereby prevention of CAA through appropriate management

## Knowledge gap

Lack of awareness of the physicians on the case definition of KD.

## Objectives of the study

- To study the clinical profile of children diagnosed as KD

- To study the patterns of coronary artery abnormalities

- To study other laboratory findings

- To study the treatment outcome

## Materials & methods

This cross-sectional (retrospective) study was done in the Department of Paediatrics in BIRDEM General Hospital from June 2019 to April 2023. A total of forty diagnosed KD patients according to the criteria of American Heart Association were included in this study [5]. After inclusion a thorough clinical evaluation were done to exclude any other problems. Patients were categorized as complete or typical KD if they had fever >5 days and ≥ 4 of the 5 of the following classic criteria of KD.

i    Erythema and cracking of lips, strawberry tongue, and/or erythema of oral and pharyngeal mucosa

ii    Bilateral bulbar conjunctival injection without exudates.

iii    Rash: maculopapular, diffuse erythema, or erythema multiforme- like.

iv    Erythema and edema of the hands and feet in acute phase and/or periungual desquamation in subacute phase

v    Cervical lymphadenopathy (≥1.5 cm diameter), usually unilateral

   Incomplete or atypical KD diagnostic criteria.

i.    Prolonged unexplained fever and 2-3/5 clinical criteria

   OR.

ii.    infants with unexplained fever 7 days AND compatible laboratory or echocardiographic findings:

   ◦ CRP ≥ 3 mg/dl or ESR ≥ 40 mm/hr, or both; + 3 or more of the following:

      - Anaemia for age

      - Platelet ≥ 4,50000

- Albumin ≤ 3 g/dl

- Elevated ALT

- Elevated WBCs ≥ 15,000/cmm

- Urine WBCs ≥ 10/hpf

◦ Z score of LAD (Left anterior descending) Coronary artery or RCA (Right Coronary artery) ≥ 2.5

◦ Or ≥ 3 other suggestive features exist, including decreased left ventricular function, mitral regurgitation, pericardial effusion or Z scores in LAD or RCA 2-2.5

Age, sex and clinical features of the children were recorded for each patient. Complete blood count (CBC), ESR, C-Reactive protein (CRP), SGPT, S. creatinine, blood C/S, Urine R/ME, C/S, echocardiography and chest X- Ray c were performed for each child and reports were recorded. All patients were treated with intravenous immunoglobulin and aspirin. Other thrombolytic agents and disease modifying drugs were used in a few cases where indicated. After improvement, patents were discharged and followed up in the Paediatric Rheumatology outdoor in BIRDEM. Follow up echocardiography was performed in all patients 6 weeks after treatment.

## Ethical statement

Approval of the waiving Ethical Review was taken from the BIRDEM Institutional Review Board. The data were accessed on November 7, 2024 and the authors had access to information that could identify individual participants during or after data collection. However, the data were deidentified before analysis of data for manuscript writing, and thus, consent from the parents was not taken.

## Sample size

Kawasaki disease (KD) exhibits significant geographic variability in incidence, ranging from 5–22 per 100,000 children <5 years in North America and Europe [6]. 4.5 per 100,000 children <15 years in India and 7–55 per 100,000 in China [8]. These findings suggest that a sample size of 33 or fewer patients is sufficient to detect meaningful differences. Since our study included 40 patients, the sample size is more than adequate.

## Statistical analysis plan

Statistical analysis of the data was done using Epi info (3.4.1). Both bi-variate and multiple logistic regression analysis were done to disclose the unadjusted and adjusted association between the clinical and laboratory parameters of complete and incomplete KD. A probability (p value) < 0.05 was considered statistically significant and the strength of association was evaluated by calculation odds ratio and their 95% confidence intervals.

## Results

Among the 40 confirmed cases of KD, 26 (65%) were diagnosed as incomplete KD and 14 (35%) as complete KD.
There were 23 (57%) males and 17 (43%) females with male to female ratio of 1.3:1. The mean age of the patients was 4.1±2.9 years.

• *S.ALT, S creatinine, S albumin, urine r/m/e & c/s, blood c/s and chest X-ray were within normal limits in all the patients* Leukocytosis: White blood cell count more than 11000/mm [10]

• Thrombocytosis: Platelet count more than 450,000/uL [11]

• Anaemia: Anemia is defined as a hemoglobin level that is two standard deviations below the mean for age [11]

- Elevated CRP > 10 mg/dL [12]

- Raised ESR: > 20 mm in 1st hr [13]

Table 1 shows that all the patients had fever and the mean duration of fever was 11.7 days. Changes in lips and oral cavity, e.g., erythema and cracking of lips, redness of tongue, and/or erythema of oral mucosa was present in 29 (72%) patients, half of the patients (50%) had polymorphous skin rash. Bilateral non-purulent conjunctival injections were noted in 18 (45%) patients, 15 (38%) had changes in distal extremities, e.g., erythema and edema of the hands and feet and periungual desquamation, 12 (30%) had unilateral cervical lymphadenopathy and erythema at BCG scars were noted in 3 (7.5%) cases. Elevated CRP in 34 (85%) patients, raised ESR in 31(78%) patients, thrombocytosis in 18(45%), anaemia in 17 (43%) and leukocytosis in 15(38%) patients.

Table 2 shows, among the forty patients initial echocardiography showed coronary artery dilatation in 33 (82.5%) patients, aneurysm in 5 (12.5%) patients and normal coronary artery in 2(5%) patients. Follow up echocardiography after 6weeks shows 27 (68%) patients had normal echocardiogram after receiving treatment, dilatation still persisted in 9(22%) cases and 4 (10%) still had aneurysm.

Table 3 shows, all the patients received IVIg and Aspirin. Four patients received warfarin along with aspirin due to persistence of coronary artery aneurysm. In one of these patients, oral steroid was also given. Two IVIg resistant cases

**Table 1. Distribution of patients according to clinical features and laboratory findings (n = 40).**

| Clinical features | Frequency (number) | Percentage (%) |
|---|---|---|
| Fever *Mean duration 11.7 days ±5.53 | 40 | 100 |
| Changes in lips and oral cavity | 29 | 72 |
| Polymorphous skin rash | 20 | 50 |
| Bilateral non-purulent conjunctival injection | 18 | 45 |
| Changes in distal extremities | 15 | 38 |
| Cervical lymphadenopathy | 12 | 30 |
| Erythema on BCG scar | 03 | 7.5 |
| Leukocytosis | 15 | 38 |
| Thrombocytosis | 18 | 45 |
| Anaemia | 17 | 43 |
| Elevated CRP | 34 | 85 |
| Raised ESR | 31 | 78 |

**Table 2. Distribution of patients according to initial echocardiography findings and after receiving treatment (n = 40).**

| Initial Echocardiography (before treatment) | | |
|---|---|---|
| Echocardiography findings | Frequency (n) | Percentage (%) |
| Normal | 2 | 5 |
| Dilatation | 33 | 82.5 |
| Aneurysm | 5 | 12.5 |
| Echocardiography findings (6 weeks after treatment) | | |
| Normal | 27 | 68 |
| Dilatation | 9 | 22 |
| Aneurysm | 4 | 10 |

**Table 3. Distribution of patients according to treatment (n = 40).**

| Name of the drugs | Frequency (n) | Percentage (%) |
|---|---|---|
| IVIg | 40 | 100 |
| Aspirin in adequate dose | 40 | 100 |
| Warfarin | 4 | 13 |
| Steroid | 3 | 8 |
| Infliximab | 1 | 3 |
| Tocilizumab | 1 | 3 |

received methyl prednisolone and one also received Infliximab. One of the patients received Tocilizumab due to concurrent KD and COVID-19 in the patient.

Table 3 shows, all the patients received IVIg and Aspirin. Four patients received warfarin along with aspirin due to persistence of coronary artery aneurysm. In one of these patients, oral steroid was also given. Two IVIg resistant cases received methyl prednisolone and one also received Infliximab. One of the patients received Tocilizumab due to concurrent KD and COVID-19 in the patient.

Number and percentages of the clinical and laboratory parameters are shown in Table 4 between complete and incomplete KD.

After adjustment of confounders in Table 5, children with complete KD more often were associated with polymorphous skin reaction and higher CRP.

Table 6 shows that Coronary artery aneurysm was significantly higher in Complete KD but there were no significant differences between Coronary artery dilatation and types of KD.

## Discussion

Kawasaki Disease is an acute systemic vasculitis that occurs predominantly in infants and young children [2]. It occurs mainly in children aged < 5 years and predominantly in male [14]. It is a self-limiting childhood illness, but it may cause coronary artery lesion and long-term morbidity [15]. Most studies have shown that percentage of typical KD is more than atypical KD [16]. However, in our study the percentage of atypical KD was more (65%). One of the hands, several studies done in Asian countries have shown that in countries particularly Japan, China, South Korea have reported higher incidence of atypical KD, partly due to genetic predisposition. Variation in environmental factors and infectious agents across the Asian countries might also influence the incidence of atypical KD in this region [17–19]. In our study, all the children

**Table 4. Clinical and laboratory findings between complete and incomplete KD.**

| | Complete KD (n = 14) Number (percentage) | Incomplete KD (n = 26) Number (percentage) |
|---|---|---|
| Changes in lips and oral cavity | 14 (100) | 15(58) |
| Polymorphous skin rash | 14 (100) | 6(23) |
| Changes in distal extremities | 13 (92) | 2(8) |
| Bilateral non-purulent conjunctival injection | 13 (92) | 5(19) |
| Cervical lymphadenopathy | 9(64) | 3(12) |
| Elevated CRP | 14(100) | 20(77) |
| Raised ESR | 12(86) | 19(73) |
| Thrombocytosis | 9(64) | 9(35) |
| Anaemia | 8(57) | 9(355 |
| Leukocytosis | 6(43) | 9(35) |

**Table 5. Bi-variate (unadjusted) and multiple logistic regression (adjusted) analysis to explore the association between complete and incomplete KD.**

| Characteristic | Unadjusted | | Adjusted | |
|---|---|---|---|---|
| | OR (95% CI) | p-value | OR (95% CI) | p-value |
| **Duration of fever** | 1.00 (0.89, 1.13) | 0.993 | | |
| **Leukocytosis** | | | | |
| No | — | | | |
| Yes | 0.71 (0.18, 2.73) | 0.608 | | |
| **Thrombocytosis** | | | | |
| No | — | | | |
| Yes | 0.29 (0.07, 1.11) | 0.078 | | |
| **Lips & redness of oral cavity** | | | | |
| No | — | | | |
| Yes | 0.12 (0.01, 0.77) | 0.060 | 0.20 (0.00, 33.8) | 0.557 |
| **Anaemia** | | | | |
| No | — | | | |
| Yes | 0.40 (0.10, 1.48) | 0.174 | | |
| **Conjunctiva** | | | | |
| No | — | | | |
| Yes | 0.05 (0.01, 0.24) | **0.001** | 0.02 (0.001, 0.21) | **0.009** |
| **High C-Reactive protein (CRP>10 mg/dl)** | | | | |
| No | — | | | |
| Yes | 0.32 (0.02, 2.31) | 0.326 | | |
| **Polymorphous skin rash** | | | | |
| No | — | | | |
| Yes | 0.03 (0.00, 0.18) | **0.002** | 0.02 (0.001, 0.44) | **0.027** |
| **Raised ESR** | | | | |
| No | — | | | |
| Yes | 0.91 (0.17, 4.20) | 0.905 | | |
| **Extremities** | | | | |
| No | — | | | |
| Yes | 0.12 (0.02, 0.51) | **0.007** | 0.23 (0.01, 3.64) | 0.309 |
| **Lymphadenopathy** | | | | |
| No | — | | — | |
| Yes | 0.14 (0.03, 0.58) | **0.009** | 0.19 (0.01, 2.23) | 0.226 |
| **Aneurysm** | | | | |
| No | — | | — | |
| Yes | 0.10 (0.00, 0.78) | 0.051 | | |
| **Dilatation** | | | | |
| No | — | | — | |
| Yes | 3.07 (0.58, 18.1) | 0.189 | | |

with KD presented with fever with average duration of around two weeks. Out of the five cardinal features, the most frequent symptom was changes in lips and oral cavity, followed by skin rash, bilateral conjunctival injection, changes in distal extremities and cervical lymphadenopathy. A study by Hua Zhu, Shao Fei Yu in Mongolia found that among the five cardinal features changes in lips (65.9%) and cervical lymphadenopathy were most common [19]. Another study done in Iran by Yazdan et al. also found that changes in lips and oral cavity were present in 70% of KD patients [17].

**Table 6. Comparison between types of KD and coronary artery abnormalities.**

**Aneurysm**

| Types of KD | Yes | No | P value |
|---|---|---|---|
| Complete | 4(28.6%) | 1(71.4%) | 0.04 |
| Incomplete | 1(3.8%) | 25(96.2%) | |

Dilatation

| Types of KD | Yes | No | P value |
|---|---|---|---|
| Complete | 10 (71.4%) | 4(28.6%) | 0.17 |
| Incomplete | 23(88.5%) | 3(11%) | |

Polymorphous skin rash and high CRP were found to be independently associated with complete KD. This result is consistent with several other studies of case series by Guida F et al. [20], Shuai S et al. [21] and Yang X et al. [15].

Cardiac lesion is the single most important complication of Kawasaki disease. In this study, 95% of the children had coronary artery abnormalities with 82.5% patients having dilatation and 12.5% patients having aneurysm. Behmadi M *et al.* found that 59% of their KD patients had coronary artery lesions [22]. In our study, all the patients except two had coronary artery abnormalities. This high percentage of cardiac involvement was most probably because by the time the patients presented to us were already in an advanced stage of the disease.

Studies have also shown that the percentage of coronary artery involvement is higher among Asian population rather than the western countries. A number of factors play an important part regarding this. The incidence of KD is significantly higher in eastern Asian countries than that of the west, therefore increasing the overall chance of CAA among the Asian population [23]. Genetic predisposition plays an important role in the development of CAA. Studies have shown several single nucleotide polymorphisms (SNPs) associated with KD in Asian populations, including genes involved in immune regulation such as ITPKC, CASP3, and FCGR2A, thus contributing to an increased inflammatory response, elevating the risk of coronary complications [24].

All the forty patients with KD were treated with IVIG, Aspirin and other thrombolytics and biological agents according to the guidelines of American Heart Association [5]. Follow up echocardiography done after 6 weeks of treatment showed that 32.5% had cardiac involvement which was previously 95% with dilatation still persisting in 9 patients and aneurysm in four patients.

In two of our IVIG resistant patients, methylprednisolone was given with one of them also received Infliximub due to persistence of fever. Infliximub, a tumour necrosis factor inhibitor is also given for the treatment of IVIG resistant cases, usually if second dose of IVIG or corticosteroid are not effective. A metanalysis study by Yan Pan, Qihong Fan and Luoyi Hu found that Infliximab exhibited remarkable antipyretic effect and they concluded that it was the best option against IVIg resistant KD [23]. One of our patients had COVID-19 and KD concurrently and was afterwards treated with Tocilizumab for COVID-19 infection. Recently published cases have shown association between Kawasaki disease and SARS COVID-2 virus which highlights the value of testing pediatric patients for COVID-19 initially diagnosed with KD [25,26].

Comparison between the clinical features and types of KD showed that all the five cardinal features were significantly more common in complete KD. Gorcyza et al. [27] and Manlhiot C et al. [28] also found similar results in their studies. We found no significant differences between most of the laboratory findings and types of KD which was also similar to the study by Manlhiot C et al. [28]. Several studies regarding coronary artery involvement between typical and atypical KD showed no significant differences between aneurysm and types of KD [28]. However, in this study it was found that coronary artery aneurysm was significantly more in typical KD.

The limitation of the study is the single centered study and provided the limited laboratory information. However, the strength of the study is the adequate sample and power of the study. We had over 90% power to detect the observed difference between groups. This enhances the accuracy and reliability of our results (e.g., narrower confidence intervals) and provides strong statistical power.

## Conclusion

This study concluded that-

- Atypical/incomplete KD were found more than the complete KD

- Boys affected more than the girls

- Ninety-five percent of KD children had coronary artery abnormalities, e.g., coronary artery dilatation and aneurysm

- Coronary artery aneurysm was detected significantly more in typical KD while Coronary artery dilatation was significantly higher in atypical KD

- Majority (95%) of the patients improved after giving IVIg

- All the 5 cardinal features were significantly higher in typical KD but no differences were noted on the haematological parameters between the types of KD.

## Recommendations

High index of suspicion is required for diagnosis of Kawasaki disease when a child presents with pyrexia of unknown origin (PUO). Echocardiography should be done in cases of suspected KD as timely intervention can prevent long term morbidity.

## Acknowledgments

We are thankful to Brigadier General Dr NNF Begum, Professor of Pediatric Cardiologist of Dhaka CMH, Bangladesh for kindly helping by performing the echocardiography of the patients.

## Author contributions

**Conceptualization:** Shareen Khan, Md Abid Hossain Mollah.

**Data curation:** Shareen Khan.

**Formal analysis:** Shareen Khan, Md Abid Hossain Mollah, Abdul Baki, Nabila Tabassum, Nasreen Islam, Amrita Lal Halder, Tasnima Ahmed, Jebun Nahar, Fauzia Mohsin, Mohammod Jobayer Chisti.

**Investigation:** Shareen Khan, Abdul Baki, Nabila Tabassum, Nasreen Islam, Amrita Lal Halder, Tasnima Ahmed, Nurun Nahar Fatema, Jebun Nahar, Fauzia Mohsin, Mohammod Jobayer Chisti.

**Methodology:** Shareen Khan, Md Abid Hossain Mollah, Abdul Baki, Nabila Tabassum, Nasreen Islam, Amrita Lal Halder, Tasnima Ahmed, Nurun Nahar Fatema, Jebun Nahar, Fauzia Mohsin, Mohammod Jobayer Chisti.

**Project administration:** Shareen Khan.

**Resources:** Shareen Khan, Abdul Baki, Nabila Tabassum, Nasreen Islam, Amrita Lal Halder, Tasnima Ahmed, Nurun Nahar Fatema, Jebun Nahar, Fauzia Mohsin, Mohammod Jobayer Chisti.

**Software:** Shareen Khan.

**Supervision:** Md Abid Hossain Mollah, Nurun Nahar Fatema, Mohammod Jobayer Chisti.

**Validation:** Mohammod Jobayer Chisti.

**Writing – original draft:** Shareen Khan.

**Writing – review & editing:** Md Abid Hossain Mollah, Abdul Baki, Nabila Tabassum, Nasreen Islam, Amrita Lal Halder, Tasnima Ahmed, Nurun Nahar Fatema, Jebun Nahar, Fauzia Mohsin, Mohammod Jobayer Chisti.

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
