## [Decision Letter · Decision Letter 0]

12 May 2025

PONE-D-24-55276Clinical & laboratory profiles and treatment outcome of Kawasaki Disease in children: Experience from a tertiary care hospitalPLOS ONE

Dear Dr. Chisti,

Thank you for submitting your manuscript to PLOS ONE. After careful consideration, we feel that it has merit but does not fully meet PLOS ONE’s publication criteria as it currently stands. Therefore, we invite you to submit a revised version of the manuscript that addresses the points raised during the review process.

We look forward to receiving your revised manuscript.

Kind regards,

Jyoti Sharma

Academic Editor

PLOS ONE

2. In the online submission form you indicate that your data is not available for proprietary reasons and have provided a contact point for accessing this data. Please note that your current contact point is a co-author on this manuscript. According to our Data Policy, the contact point must not be an author on the manuscript and must be an institutional contact, ideally not an individual. Please revise your data statement to a non-author institutional point of contact, such as a data access or ethics committee, and send this to us via return email. Please also include contact information for the third party organization, and please include the full citation of where the data can be found.

Reviewers' comments:

Reviewer's Responses to Questions

**Comments to the Author**

1. Is the manuscript technically sound, and do the data support the conclusions?

Reviewer #1: Partly

Reviewer #2: Yes

2. Has the statistical analysis been performed appropriately and rigorously? 

Reviewer #1: No

Reviewer #2: Yes

3. Have the authors made all data underlying the findings in their manuscript fully available?

Reviewer #1: Yes

Reviewer #2: Yes

4. Is the manuscript presented in an intelligible fashion and written in standard English?

Reviewer #1: Yes

Reviewer #2: Yes

5. Review Comments to the Author

Reviewer #1: This manuscript examines Kawasaki Disease in children at a Bangladeshi tertiary care hospital, providing valuable data through detailed clinical and laboratory profiles of 40 cases. While the study's strengths include clear documentation of treatment outcomes, several concerns need addressing - notably the unusually high rates of incomplete KD and coronary abnormalities, statistical rigor, methodology clarity, and limited discussion of findings in context of current literature. With appropriate revisions, this work could meaningfully contribute to understanding KD in Bangladesh.

Major Comments:

1. While several laboratory tests were performed (CBC, ESR, CRP, SGPT, S. creatinine, blood C/S, Urine R/ME, C/S), the Results section only presents and analyzes a limited subset of these findings, primarily focusing on CRP, ESR, and CBC parameters. The data from other tests like SGPT, creatinine, blood culture, and urinalysis are mentioned in Methods but not reported or analyzed in the Results. To maintain scientific rigor and completeness, either all laboratory data should be presented and analyzed to justify the term "laboratory profiles" in the title, or the title should be modified to reflect the actual scope of reported findings.

2. Your introduction requires updating with literature from 2022-2024 to better contextualize your findings within current Kawasaki Disease research. We recommend incorporating recent epidemiological data (especially from South Asia), modern diagnostic approaches, and contemporary treatment outcomes. While retaining seminal papers, please replace pre-2015 references with current evidence on regional epidemiology, diagnostic criteria, and treatment guidelines to strengthen your manuscript's relevance and contribution to the field.

3. Clarify diagnostic criteria used for incomplete KD.

4. The statistical methodology requires significant revision. While P-values are presented, the specific statistical tests used are not specified, and there is no sample size calculation or power analysis. The authors should clearly state which statistical tests were employed and provide justification for the sample size to ensure scientific validity.

5. Add multivariate analysis of risk factors for coronary involvement.

6. Figure 1 is overly simple - additional figures showing key findings would be valuable.

7. The Discussion section needs substantial revision as it currently focuses too heavily on restating results rather than interpretation and context. We recommend reducing the descriptive content of findings and expanding the analytical discussion, particularly comparing your results with contemporary literature and explaining the high rates of incomplete KD and coronary involvement observed in your study. Include more critical analysis of how your findings contribute to current understanding of Kawasaki Disease in South Asian populations.

8. The authors should strengthen the discussion by citing contemporary literature to explain why their study found unusually high rates of incomplete Kawasaki Disease (65%) and coronary involvement (95%) compared to global data, with particular focus on whether these findings reflect delayed diagnosis, regional genetic factors, or referral bias in their tertiary care setting.

Reviewer #2: 1. The dose of Aspirin in the acute phase may also affect the coronary artery on follow-up. It needs to be kept under consideration and can be included if it varies.

2. The article is an addition to the previous studies where atypical Kawasaki is increasing now.

6. PLOS authors have the option to publish the peer review history of their article (what does this mean? ). If published, this will include your full peer review and any attached files.

**Do you want your identity to be public for this peer review?** For information about this choice, including consent withdrawal, please see our Privacy Policy .

Reviewer #1: **Yes: ** Zhenglin chang

Reviewer #2: **Yes: ** Dr.Gunjan Baweja

---

## [Author Response · Author response to Decision Letter 1]

29 Jun 2025

Responses to the comments of the academic editor

Comments to the Author

1. Is the manuscript technically sound, and do the data support the conclusions?

Reviewer #1: Partly

Reviewer #2: Yes

Response: We have now revised the manuscript to make this technically sound and the data now support the conclusion

2. Has the statistical analysis been performed appropriately and rigorously?

Reviewer #1: No

Reviewer #2: Yes

Response: Thanks. We have revised the statistical analysis section to make this more appropriate and rigorous ________________________________________

3. Have the authors made all data underlying the findings in their manuscript fully available?

Reviewer #1: Yes

Reviewer #2: Yes

Response: Thanks. ________________________________________

4. Is the manuscript presented in an intelligible fashion and written in standard English?

Reviewer #1: Yes

Reviewer #2: Yes

Response: Thanks.________________________________________

5. Review Comments to the Author

Reviewer #1: This manuscript examines Kawasaki Disease in children at a Bangladeshi tertiary care hospital, providing valuable data through detailed clinical and laboratory profiles of 40 cases. While the study's strengths include clear documentation of treatment outcomes, several concerns need addressing - notably the unusually high rates of incomplete KD and coronary abnormalities, statistical rigor, methodology clarity, and limited discussion of findings in context of current literature. With appropriate revisions, this work could meaningfully contribute to understanding KD in Bangladesh.

Response: Thanks for your constructive review comments. We have revised the manuscript following your kind suggestions.

Major Comments:

1. While several laboratory tests were performed (CBC, ESR, CRP, SGPT, S. creatinine, blood C/S, Urine R/ME, C/S), the Results section only presents and analyzes a limited subset of these findings, primarily focusing on CRP, ESR, and CBC parameters. The data from other tests like SGPT, creatinine, blood culture, and urinalysis are mentioned in Methods but not reported or analyzed in the Results. To maintain scientific rigor and completeness, either all laboratory data should be presented and analyzed to justify the term "laboratory profiles" in the title, or the title should be modified to reflect the actual scope of reported findings.

Response: Thanks. We have provided the information of all mentioned investigations in the results (page 6 under table 1).

2. Your introduction requires updating with literature from 2022-2024 to better contextualize your findings within current Kawasaki Disease research. We recommend incorporating recent epidemiological data (especially from South Asia), modern diagnostic approaches, and contemporary treatment outcomes. While retaining seminal papers, please replace pre-2015 references with current evidence on regional epidemiology, diagnostic criteria, and treatment guidelines to strengthen your manuscript's relevance and contribution to the field.

Response: Thanks. We have revised the introduction section according to your kind suggestions on pages 3 & 3 under introduction.

3. Clarify diagnostic criteria used for incomplete KD.

Response: Thanks. It is now mentioned on pages 4 and 5 under section material and methods section.

4. The statistical methodology requires significant revision. While P-values are presented, the specific statistical tests used are not specified, and there is no sample size calculation or power analysis. The authors should clearly state which statistical tests were employed and provide justification for the sample size to ensure scientific validity.

Response: Thanks. This is an exploratory study where we enrolled all the children (n=40) diagnosed with KD between June 2019 to April 2023. However, we have performed the sample size calculation which has been mentioned on page under sample size on page 5. We have also done the power analysis of the study which is discussed on page 14, prior conclusion, under discussion section. Moreover, we have revised the statistical analysis plan according to the suggestion of the respected reviewer.

5. Add multivariate analysis of risk factors for coronary involvement.

Response: Thanks. We have introduced multiple logistic regression and mentioned this in the statistical analysis plan on page 6 (first paragraph) and results section (table V, page 10 on results section)

6. Figure 1 is overly simple - additional figures showing key findings would be valuable.

Response: Thanks. We have deleted this figure as this information is available in the results on page 6.

7. The Discussion section needs substantial revision as it currently focuses too heavily on restating results rather than interpretation and context. We recommend reducing the descriptive content of findings and expanding the analytical discussion, particularly comparing your results with contemporary literature and explaining the high rates of incomplete KD and coronary involvement observed in your study. Include more critical analysis of how your findings contribute to current understanding of Kawasaki Disease in South Asian populations.

Response: Thanks. We have revised discussion section on pages 11 to 13 addressing the above-mentioned suggestions.

8. The authors should strengthen the discussion by citing contemporary literature to explain why their study found unusually high rates of incomplete Kawasaki Disease (65%) and coronary involvement (95%) compared to global data, with particular focus on whether these findings reflect delayed diagnosis, regional genetic factors, or referral bias in their tertiary care setting.

Response: Thanks. We have done this accordingly throughout the manuscript and also in the reference list.

Reviewer #2: 1. The dose of Aspirin in the acute phase may also affect the coronary artery on follow-up. It needs to be kept under consideration and can be included if it varies.

Response: Thanks. We concur with the concern of the respected reviewer and. It is important to mention that we have used adequate dose of aspirin and mentioned in table III.

2. The article is an addition to the previous studies where atypical Kawasaki is increasing now.

Response: Many thanks for your positive comments.

---

## [Editor Report · Decision Letter 1]

2 Jul 2025

Clinical & laboratory profiles and treatment outcome of Kawasaki Disease in children: Experience from a tertiary care hospital

PONE-D-24-55276R1

Dear Dr.Chisti,

We’re pleased to inform you that your manuscript has been judged scientifically suitable for publication and will be formally accepted for publication once it meets all outstanding technical requirements.

Kind regards,

Jyoti Sharma

Academic Editor

PLOS ONE

Additional Editor Comments (optional):

Thank you for thoughtfully addressing the reviewers' suggestions. The revisions have significantly enhanced the quality and clarity of the manuscript